# The Description and Analysis of the Complete Genome of *Dermacoccus barathri* FBCC-B549 Strain

**DOI:** 10.3390/microorganisms12061227

**Published:** 2024-06-18

**Authors:** Yeha Kim, Hyaekang Kim, Jina Kim, Ji-Hye Han, Eu Jin Chung, Seung Won Nam, Miyoung Shin, Woori Kwak

**Affiliations:** 1Department of Medical and Biological Sciences, The Catholic University of Korea, Bucheon 14662, Republic of Korea; 2Department of Biotechnology, The Catholic University of Korea, Bucheon 14662, Republic of Korea; 3Bio-Resources Bank Division, Nakdonggang National Institute of Biological Resources, Sangju 37242, Republic of Korea; 4Performance Innovation Division, Nakdonggang National Institute of Biological Resources, Sangju 37242, Republic of Korea; 5Department of Pathology, Yale University School of Medicine, New Haven, CT 06510, USA

**Keywords:** complete genome assembly, *Dermacoccus barathri*, comparative analysis, potential pathogenicity, biofilm formation

## Abstract

*Dermacoccus barathri* is the first reported pathogen within the *Dermacoccus* genus to cause a catheter-related bloodstream infection, which occurred in 2015. In this study, the complete genome assembly of *Dermacoccus barathri* was constructed, and the complete genome of *Dermacoccus barathri* FBCC-B549 consists of a single chromosome (3,137,745 bp) without plasmids. The constructed genome of *D. barathri* was compared with those of two closely related species within the *Dermacoccus* genus. *D. barathri* exhibited a pattern similar to *Dermacoccus abyssi* in terms of gene clusters and synteny analysis. Contrary to previous studies, biosynthetic gene cluster (BGC) analysis for predicting secondary metabolites revealed the presence of the LAP biosynthesis pathway in the complete genome of *D. barathri*, predicting the potential synthesis of the secondary metabolite plantazolicin. Furthermore, an analysis to investigate the potential pathogenicity of *D. barathri* did not reveal any antibiotic resistance genes; however, nine virulence factors were identified in the Virulence Factor Database (VFDB). According to these matching results in the VFDB, despite identifying a few factors involved in biofilm formation, further research is required to determine the actual impact of *D. barathri* on pathogenicity. The complete genome of *D. barathri* is expected to serve as a valuable resource for future studies on *D. barathri*, which currently lack sufficient genomic sequence information.

## 1. Introduction

The *Dermacoccus* genus comprises coccoid-shaped bacteria characterized as Gram-positive, lacking capsules, non-spore-forming, aerobic, and commonly found in soil, water, and cured meat products [1]. In addition, some strains are part of the human skin microbiome, and their decrease has been associated with Atopic dermatitis, according to Wongpiyabovorn et al. in 2019 [2]. Despite inhabiting a wide range of terrestrial, aquatic, and biological habitats, only four fully identified species are known within the *Dermacoccus* genus: *Dermacoccus barathri*, *Dermacoccus abyssi*, *Dermacoccus nishinomiyaensis*, and *Dermacoccus profundi* [3,4]. *Dermacoccus* species have traditionally been considered nonpathogenic commensals that live on human skin. However, *D. barathri* was identified as the first potential pathogen in *Dermacoccus* species because it has been implicated in catheter-related bloodstream infections (CRBSIs) [5]. Beginning with *D. barathri*, a few species of the *Dermacoccus* genus have caused human infections. Specifically, there have been five reported cases of infections caused by *D. nishinomiyaensis* [6,7,8,9,10] and one reported case of infection caused by *Dermacoccus* sp. [11].

CRBSIs often result from bacterial colonization near the catheter insertion site, where biofilm formation protects bacteria from antibiotics and immune responses [12]. Gram-positive bacteria, including *D. barathri*, are more frequently associated with catheter infections than Gram-negative bacteria [12]. Notably, *D. barathri* was reported in 2015 as the first *Dermacoccus* species to cause a CRBSI in a 7-year-old boy with an implantable central venous access device [5]. Furthermore, *D. barathri* isolated from this patient showed the ability to form biofilms after approximately 10 days of culture [5]. Despite being recognized as a novel pathogen within the *Dermacoccus* genus, the complete genome of and genetic information on *D. barathri* are lacking. Therefore, constructing the complete genome of *D. barathri* can address this limitation and aid in further research.

In this study, whole-genome sequencing of *Dermacoccus barathri* FBCC-B549 was conducted using Nanopore Flongle, resulting in the assembly of the complete genome of *D. barathri*. Through comparative genomic analysis with two closely related species, *D. abyssi*, and *D. nishinomiyaensis*, the evolutionary relationships and unique genetic features of *D. barathri* were elucidated. In addition, genes associated with antibiotic resistance and virulence factors, particularly those linked to biofilm formation, were scrutinized to evaluate the potential pathogenicity of *D. barathri*. Furthermore, secondary metabolites that had not been previously discovered before constructing the complete genome were identified in this study. The comprehensive analysis of the complete genome of *D. barathri* sheds light on its genetic characteristics and potential pathogenic traits. This study promises to enrich our molecular-level understanding and serve as a valuable resource for further research in related fields.

## 2. Materials and Methods

### 2.1. Whole-Genome Sequencing

The DNA of *D. barathri* FBCC-B549 was extracted using the Omega Bio Mag-Bind Universal Pathogen Kit (Omega Bio-tek, Norcress, GA, USA) following the manufacturer’s protocol. The Nanopore Flongle sequencing library was constructed using a Nanopore SQK-LSK114 kit (Oxford Nanopore Technologies, Oxford, UK). Base calling for the generated signal from the nanopore was conducted using Guppy v6.5.7 [13] with a super-accuracy model and CUDA acceleration. Trimming of adapter sequences in the generated reads was performed using Porechop_ABI v0.5.0 [14], and the trimmed reads were assembled using FLYE v2.9.1 [15] with the nano-hq parameter. For a more accurate genome sequence, Medaka v1.8.0 (https://github.com/nanoporetech/medaka, accessed on 14 August 2023) was employed for polishing with a super-accuracy calling model. The quality of the assembled genome was evaluated using BUSCO v5.4.7 [16] with the micrococcales_odb10 database.

### 2.2. Gene prediction and Functional Annotation

The gene annotation and characteristics of *D. barathri* FBCC-B549 were analyzed using Prokka, COGclassifier, and the Rapid Annotation based on Subsystem Technology (RAST) server. Prokka v1.14.5 [17] and Proksee [18] were utilized for gene prediction, annotation, and the construction of a circular map of the assembled genome. COGclassifier v1.0.5 [19] was employed to annotate genes and classify them according to COG. RAST [20] was also utilized for gene annotation and subsystem category-wise analyses.

### 2.3. Comparative Genome Analysis

Comparative genomic analyses were conducted using the complete genomes of closely related species within the *Dermacoccus* genus, namely, *D. abyssi* and *D. nishinomiyaensis*, which are the species that have available complete genomes. Whole-genome comparisons were performed using Mauve v2.4.0 [21]. To align the starting point of the circular genome, each genome of *D. barathri*, *D. abyssi*, and *D. nishinomiyaensis* was manually modified. The locally collinear block (LCB) weight was set to 1625. Gene clusters of the three species were compared and annotated using OrthoVenn3 [22] with default parameters (E-value 0.01; inflation value 1.50). Additional comparative genome analyses were performed using Roary v3.13.0 [23]. Biosynthetic gene clusters (BGCs) were predicted using antiSMASH v7.1.0 [24] with default parameters.

### 2.4. Prediction of Antimicrobial Resistance Gene and Potential Virulence Factors

The potentially harmful antibiotic resistance and virulence of *D. barathri* were predicted. Resfinder v4.4.2 [25] and AMRFinderPlus v3.11.26 [26] were used to identify antibiotic resistance factors using the default parameters. Potential virulence prediction and toxin factor analysis were performed against the VFDB [27] using the DIAMOND v2.1.8 tool [28]. Toxicity factors were filtered under relaxed conditions with an identity above 70% and query coverage above 60% [29].

## 3. Results and Discussion

### 3.1. Complete Genome Construction and Gene Annotation

Using Nanopore Flongle sequencing data, the complete genome of *D. barathri* was successfully assembled using Flye and Medaka. The genome of *D. barathri* is composed of a single chromosome with no plasmids, totaling 3,137,745 base pairs and possessing a GC content of 68.2%, which is typical for Actinomycetes (Table 1). High-GC regions are prone to PCR bias, resulting in reduced amplification efficiency under standard PCR conditions. This bias likely contributed to the challenges in obtaining the complete genome of *D. barathri* using Illumina sequencing, which relies on PCR. However, nanopore sequencing, based on single-molecule sequencing, is believed to be less affected by PCR bias, facilitating the assembly of the high-GC-content genome of *D. barathri*. To evaluate the quality of the assembled genome, BUSCO was used with the micrococcales_odb10 database, revealing a high level of completeness at 98.5%. This included 528 complete and single-copy BUSCOs, one complete and duplicated BUSCO, and eight missing BUSCOs (Table 2). Previous studies have suggested that nanopore sequencing can solely produce qualified genomes without additional short-read data for specific research purposes, and the polishing step is critical [30]. While tools like Homopolish [31] have been effective in enhancing the base accuracy of nanopore sequencing by utilizing multiple public genomes of conspecific microorganisms, this specific polishing tool could not be utilized for *D. barathri* due to the limited availability of sequences in the public database. Nevertheless, the accuracy enhancement achieved using nanopore protein pores, associated reagents, base calling (super-accuracy model), and polishing algorithms, particularly with the default Medaka polishing tool provided by Nanopore, was deemed sufficient through BUSCO analysis. As technology advances, it is increasingly expected that the construction of qualified genomes suitable for various research purposes will become more feasible solely through the use of nanopore technology.

For the gene annotation of the complete genome, Prokka annotation was performed, identifying 2924 CDSs, 56 tRNAs, and 9 rRNAs (Table 1). These values are similar to those of the closely related species *D. abyssi* and *D. nishinomiyaensis*. A circular genome map was constructed based on the gene information of the complete *D. barathri* genome using Proksee (Figure 1).

Additionally, functional annotation of the genes was conducted using the COG classifier and RAST annotation. In the COG classifier results, COG annotation was completed for 2256 of the 2924 CDS (77.15%). Categories related to basic nutrient metabolism, such as “Translation, Ribosomal structure, and biogenesis” (212), “Amino Acid Transport and Metabolism” (209), “Carbohydrate Transport and Metabolism” (171), “Lipid Transport and Metabolism” (152), and “Coenzyme Transport and Metabolism” (152), accounted for a significant portion (Figure 2). Considering the requirement for the secretion of polysaccharides, proteins, amino acids, and various RNAs and DNAs outside the cell for biofilm matrix formation, these basic nutrient-metabolism-related categories are expected to play important roles in biofilm formation. In addition to COG annotation, subsystem-based annotation using the RAST server was performed, with 25% of the annotated genes assigned to subsystems and 75% unassigned. Similar to the COG annotation results, a considerable number of genes involved in basic metabolic processes, such as “Amino Acids and Derivatives” (228), “Protein Metabolism” (170), and “Carbohydrates” (138), were identified to be part of subsystems (Figure 2). Furthermore, through RAST, 29 genes were identified in the “Virulence, Disease, and Defense” category, with 20 genes classified under the “Resistance to Antibiotics and Toxic Compounds” subcategory and 9 genes under the “Invasion and Intracellular Resistance” subcategory. This suggests that the species possesses genes that may contribute to its pathogenicity.

### 3.2. Comparative Genome Analysis

Using Mauve, a comparison of the LCBs within the genomes of the closely related species *D. abyssi* and *D. nishinomiyaensis* was conducted (Figure 3). The regions depicted as blocks in Figure 3 represent LCBs, denoting conserved sequence regions devoid of internal genomic rearrangements. The overall GC content and genome length of the three species were similar, and most gene sequences and orders were preserved in the major LCB arrays, indicating good collinearity. While the major LCB arrays of the three species were similar, the crossing patterns of the lines linking the conserved regions in each genome appeared to be more complex in *D. nishinomiyaensis*, suggesting a closer similarity to *D. abyssi* in terms of overall genome structure. According to the phylogenetic tree of the *Dermaoccus* genus reported in the literature, *D. barathri* is phylogenetically closer to *D. abyssi* than *D. nishinomiyaensis* [1].

This phylogenetic relationship is consistent with the results reported by Roary. *D. barathri* shared 2176 genes with *D. abyssi* and 38 with *D. nishinomiyaensis*. Analyses using OrthoVenn3 showed a similar pattern (Figure 4). As illustrated in the Venn diagram in Figure 4, the total number of proteins in *D. barathri*, *D. abyssi*, and *D. nishinomiyaensis* was 8444, with 2679 clusters. Furthermore, 2056 clusters were shared among the three species. These clusters constituted a significant proportion of individual clusters and primarily consisted of genes related to fundamental bacterial biological functions. Similar to previous results, *D. barathri* shared more clusters with *D. abyssi*, sharing 2469 clusters. *D. barathri* shared 2170 clusters with *D. nishinomiyaensis*, which was fewer than the number of clusters shared by *D. abyssi*. Although the patterns between the results of Roary and OrthoVenn3 appear similar, variations in the values suggest differences in the default parameters for each tool.

By predicting secondary metabolites using antiSMASH, biosynthetic gene clusters (BGCs) in *D. barathri* and related species were investigated. In the complete genome of *D. barathri* FBCC-B549, six BGCs for secondary metabolites were identified: the NI-siderophore, NAPAA, terpene, RiPP-like, LAP, and ectoine. Among these, the NAPAA, terpene, and RiPP-like clusters were found to be common to *D. nishinomiyaensis*, *D. abyssi*, and *D. barathri*. Additionally, *D. abyssi* and *D. barathri* shared two additional clusters: the NI-siderophore and ectoine. In contrast to previous studies [32], the analysis of the newly constructed complete genome of *D. barathri* FBCC-B549 using antiSMASH revealed the presence of the LAP (Linear azol(in)e-containing peptide) cluster, which was not previously reported in *D. barathri* literature. LAPs belong to a specific subgroup of ribosomally synthesized and post-translationally modified peptides (RiPPs) characterized by the presence of azole or azoline heterocycles within the linear peptides [33]. RiPPs and LAPs undergo pathways in which precursor peptides mature and become active through the actions of modifying enzymes and transporters [33]. Among these modifying enzymes, YcaO serves as a biosynthetic enzyme that forms azole and azoline heterocycles [33]. The LAP cluster in *D. barathri* also contains the YcaO protein gene and shares the same precursor peptide sequence with that of *Dermacoccus* sp. DE3. Previous studies suggested that *Dermacoccus* sp. DE3 synthesizes plantazolicin (PZN) as a secondary metabolite [33], indicating the potential for *D. barathri* to also synthesize PZN. PZN was initially discovered in the plant growth-promoting bacterium *Bacillus amyloliquefaciens* FZB42 [34]. PZN acts as a ribosome-targeting inhibitor and inhibits translation by binding to bacterial 70S ribosomes. It has been reported to function as an antibiotic with a very narrow spectrum, targeting *Bacillus anthracis*, the causative agent of anthrax [33,35]. Therefore, *D. barathri* appears to have the potential to synthesize antimicrobial substances effective against anthrax-causing bacteria. This characteristic is expected to be further explored in subsequent research.

### 3.3. Potential Pathogenicity of D. barathri

To explore the potential pathogenicity of *D. barathri* (FBCC-B549), an analysis of antibiotic resistance genes and virulence factors within its complete genome was conducted. Unfortunately, our initial attempt to identify antimicrobial resistance genes using Resfinder and AMRfinder with default parameters did not yield matches to known antibiotic resistance genes. Subsequently, for virulence factor analysis, protein sequences obtained from the Prokka annotation of the complete genome of *D. barathri* FBCC-B549 were employed and compared against VFDB using DIAMOND. Our criteria for matches were set at identity ≥70% and query coverage ≥60%, in line with established parameters from previous research [29]. Through this analysis, nine virulence factors were identified and categorized into nutritional/metabolic factors, regulation, adherence, stress survival, and others (Table 3). As previously documented, in the first case of a catheter-related bloodstream infection, *D. barathri* was found to possess the ability to form biofilms [5]. Assuming this trait to be common among *D. barathri* strains, potential factors contributing to biofilm formation among the detected virulence factors were sought. Factors within the adherence category, such as GroEL and EF-Tu, are known for their involvement in the initial adhesion phase of biofilm formation across various pathogens [36]. EF-Tu serves as a prominent constituent of bacterial cell walls, exhibiting properties similar to those of adhesion factors, inducing inflammatory responses, and participating in energy metabolism, functioning similarly to moonlighting proteins [37]. Moreover, EF-Tu plays a role similar to that of amyloid-like proteins discovered in *Gallibacterium* during biofilm formation [38], contributing to immune evasion, initiation, maturation, and attachment stages of biofilm development [39]. Furthermore, GroEL participates in early aggregation [40], and beyond its essential role in mycolate synthesis during *Mycobacterium* biofilm maturation [41], it has also been implicated in biofilm formation in *Hemophilus influenzae* [35]. LeuD, present in the nutritional/metabolic factor category, was upregulated in *E. coli* biofilms, and experimental evidence showed that LeuD functions as a major regulator of biofilm growth as it becomes S-nitrosylated [42,43]. Finally, KatA, within the stress survival category, safeguards biofilms and organisms from UVA radiation [44]. In addition to these biofilm-related factors, *aceA*, a gene encoding isocitrate lyase, regulates the TCA cycle and synthesizes carbohydrates from fatty acids via the initial enzyme of the glyoxylate shunt, contributing to sustaining infection; therefore, it is classified as a virulent factor [45]. Additionally, SigA, SigH, and RegX3 within the regulation category are known to regulate virulence and stress resistance operons [26,46,47]. Biofilm formation is associated not only with host immune responses and antibiotic therapy but also with the recurrence and persistence of chronic diseases. This also suggests its involvement in the pathogenicity of opportunistic bacteria [48]. According to the National Institutes of Health, up to 80% of human bacterial infections are associated with biofilm-associated microorganisms [49,50,51]. *D. barathri* has also been reported to form biofilms and cause catheter-related bloodstream infections [4]. Additionally, despite not being extracted from the reported case, the strain used in this study possessed genetic factors that are potentially involved in biofilm formation. Further research is required to ascertain the biofilm-forming capabilities of these genes. Furthermore, subsequent studies should investigate whether biofilm formation is a common phenotypic trait in *D. barathri*. The complete genome of *D. barathri* constructed in this study is expected to serve as a valuable resource for enhancing our understanding of the characteristics and genetic background of *D. barathri* species lacking an available genome.

## 4. Conclusions

In this study, the complete genome of *D. barathri* (strain FBCC-B549) was successfully assembled using Nanopore Flongle technology, thereby filling a significant gap in the available public genomic information for this species. The genome of *D. barathri* has a notably high GC content, and our assembly reveals a single chromosome spanning 3,137,745 base pairs. From this complete genome, fundamental genetic information was extracted, and genes involved in secondary metabolite synthesis were identified. These findings were then compared with sequences from other species within the same genus. Furthermore, antibiotic resistance genes and virulence factors were analyzed to evaluate the potential pathogenicity of *D. barathri* at the genetic level. The establishment of the complete genome sequence of *D. barathri* in this study is expected to significantly advance our molecular-level understanding of this microbial species. Moreover, it will serve as a valuable resource for further research in related fields.

## Figures and Tables

**Figure 1 microorganisms-12-01227-f001:**
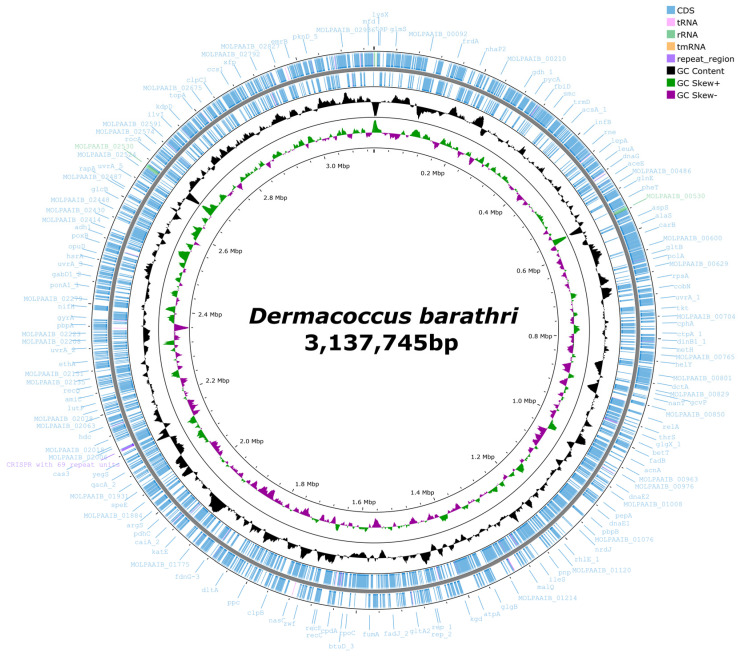
A circular genome map of *D. barathri* generated using Prokka and Proksee, illustrating the locations of coding sequences (CDSs) within the genome.

**Figure 2 microorganisms-12-01227-f002:**
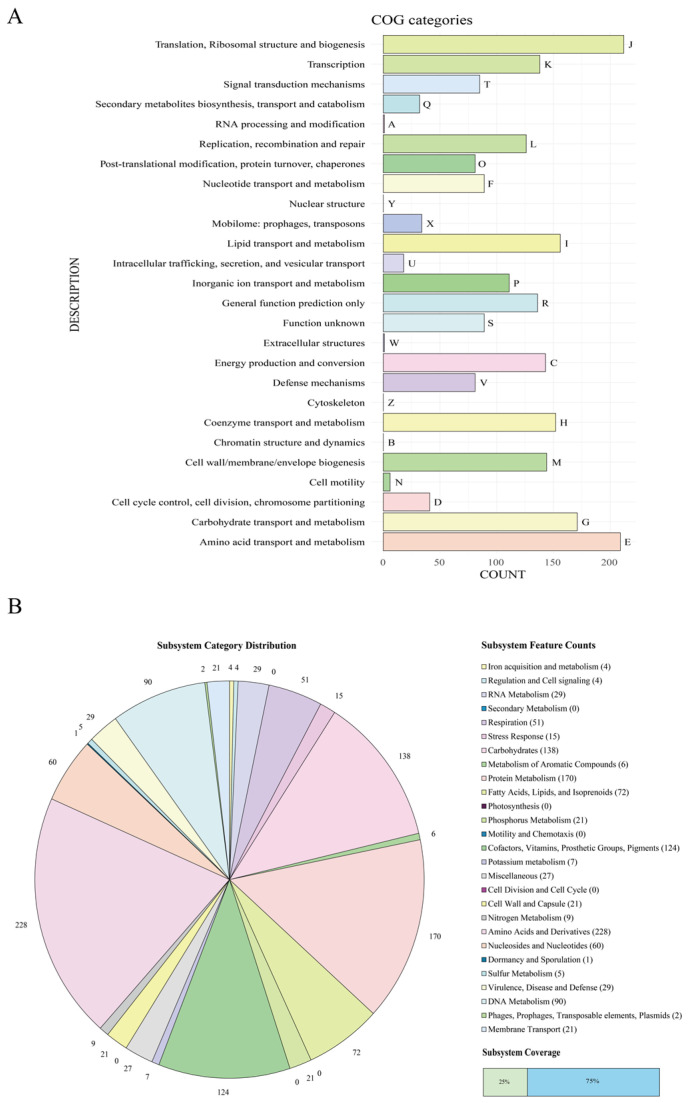
(**A**) COG annotation results for *D. barathri*, with each letter representing a class of COG categories. (**B**) RAST annotation results for *D. barathri*, with the pie chart displaying Subsystem Feature Counts and the number of genes associated with specific functions.

**Figure 3 microorganisms-12-01227-f003:**
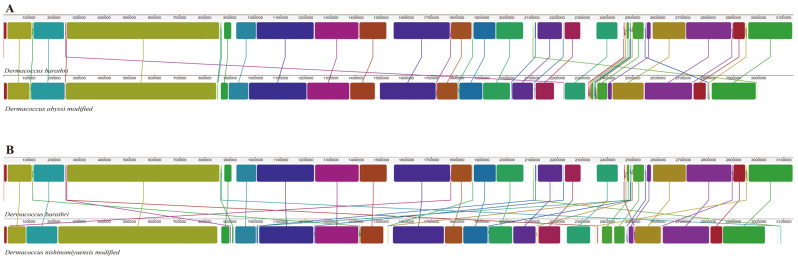
Locally collinear block (LCB) comparison among complete genomes of *D. barathri*, *D. abyssi* (**A**), and *D. nishinomiyaensis* (**B**). Colored blocks indicate aligned genome sequences, highlighting homologous and conserved regions.

**Figure 4 microorganisms-12-01227-f004:**
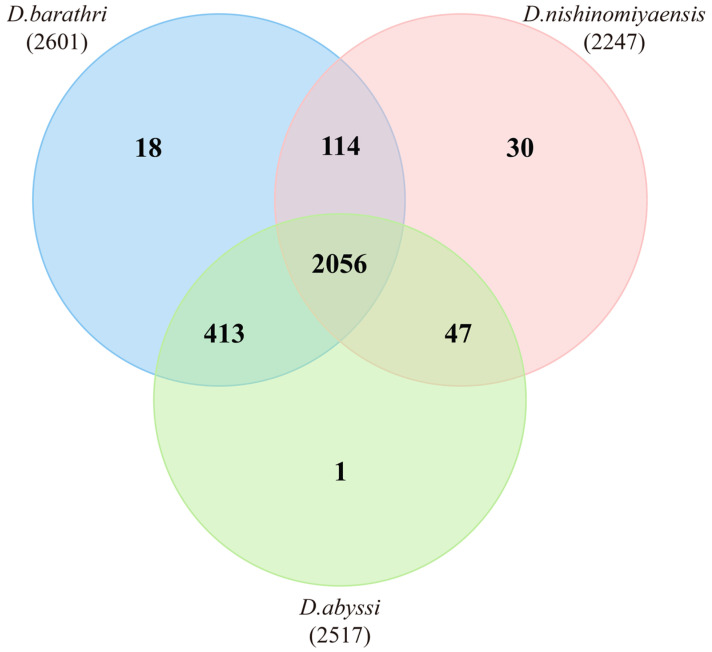
Orthologous gene cluster analysis using OrthoVenn3, depicting shared orthologous clusters among three *Dermacoccus* species. Numbers adjacent to species indicate the total clusters in each list.

**Table 1 microorganisms-12-01227-t001:** Summary of assembly statistics for the *D. barathri* FBCC-B549 whole genome. “bp” means “base pair”, which is the unit used to measure the length of DNA.

Species Name	*Dermacoccus barathri* FBCC-B549
NCBI taxon ID	322601
Taxonomy	Bacteria; Actinomycetota; Actinomycetes; Micrococcales; Dermacoccaceae; Dermacoccus
Genome Size (bp)	3,137,745
GC content in the DNA	68.2 mol% G+C
Number of Genome Sequences	1 Circular
Number of Plasmids	0
Number of Coding Sequences	2924
Number of rRNAs	9
Number of tRNAs	56

**Table 2 microorganisms-12-01227-t002:** Summary of BUSCO analysis for *D. barathri* genome.

Categories	No. Genes	Percentage
Complete	529	98.5
Complete single-copy BUSCOs	528	98.3
Complete duplicate BUSCOs	1	0.2
Fragmented BUSCOs	0	0
Missing BUSCOs	8	1.5

**Table 3 microorganisms-12-01227-t003:** Virulence factors identified in *D. barathri* using blastp with the VFDB database.

Category	Virulence Factors
Nutritional/metabolic factors	LeuD, SugC
Regulation	SigA, SigH, RegX3
Adherence	EF-Tu, GroEL
Stress survival	KatA
Others	AceA

## Data Availability

The original data presented in the study are openly available in the NCBI database Bioproject under accession number PRJNA1118424.

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
