# Peer review of "The Description and Analysis of the Complete Genome of Dermacoccus barathri FBCC-B549 Strain"

_microorganisms, 2024, doi:10.3390/microorganisms12061227_

Round 1

Reviewer 1 Report

Comments and Suggestions for Authors

The manuscript “First Complete Genome and Comparative Analysis of Dermacoccus barathri” aimed to describe and analyze the genome of Dermacoccus barathri FBCC-B549 strain. The research is interesting and brings important information to the literature regarding Dermacoccus barathri species.

General comments:

The first person singular or plural is not appropriate in scientific language. Rewrite the text impersonally (L19, L59, L64, L79, L143, L231, L242, L274, L276, L282, L293, L294, L315).

Readers place very low value on claims of primacy ("This is the first case of ...," "To our knowledge, this is the first study to…," etc.") Avoid including any such claims in your manuscript.

Title: Suggestion to change to: “Description and analysis of the complete of Dermacoccus barathri FBCC-B549 strain”

Abstract:

L26 – Describe the abbreviation VFDV

1. Introduction

L35 and L51 – Correct: “Gram”

L42 – The correct here is also to cite the List of Prokaryotic names with Standing in Nomenclature (LPSN)

Parte, A.C., Sardà Carbasse, J., Meier-Kolthoff, J.P., Reimer, L.C. and Göker, M. (2020). List of Prokaryotic names with Standing in Nomenclature (LPSN) moves to the DSMZ. International Journal of Systematic and Evolutionary Microbiology, 70, 5607-5612; DOI: 10.1099/ijsem.0.004332

L58-67 - This paragraph is describing many information’s that seems more “Material and Methods” section. Please rewrite with the importance of the study and its main aim.

2. Material and methods

General comments:

The brand, city and country of manufacturing of all reagents and equipment’s used in the study must be provided.

L89-91 - Why did the authors not use the genome of the specie Dermacoccus profundi for comparative genome analysis? Is it not available? If not, this information must be provided in the text.

L104-105 – What is the reference for these % of identity and coverage?

L111 – Cite the “Table 1” in the end of the paragraph.

3. Results and Discussion

Table 1 – Include the description of the abbreviation “bp” in the Table legend.

L177 – Put D. barathri, D. abyssi, and D. nishinomiyaensis in italic form.

L194 – Put Dermacoccus in italic form.

L218 - Put D. barathri in italic form.

L230 – Change “9” to “nine”

L247 – Put E. coli in italic form.

L257-259 – The reference “Römling, U.; Balsalobre, C. Biofilm infections, their resilience to therapy and innovative treatment strategies. Journal of internal medicine 2012, 272, 541-561”is old (>5 years). Does this percentage of over 80% still stand today? Please up-to-date.

Reviewer 2 Report

Comments and Suggestions for Authors

The manuscript entitled: “First Complete Genome and Comparative Analysis of Dermacoccus barathri”, the manuscript is well written well discussed and provides, in my point of view, interesting and useful insights.

Nevertheless, the authors should carefully and thoroughly analyse the formatting of taxonomy. Namely, in line 41, the abbreviated name of the species should not be defined as other common acronyms. The species names, in their first appearance must be displayed in full (in the abstract and in the body text), for example: Dermacoccus barathri. In the second appearance and in the following the abbreviation of its genus should be used: D. barathri. Please carefully revise the entire manuscript. Furthermore, I found several genera and specific epithets not italicized. Please carefully revise the entire manuscript.

Minor comments:

The culture conditions prior to genome extraction would be, in my point of view,

Table 1 taxonomy, please revise, not all the terms require to be italicized

Figure 2, A, please revise the typo on post-translational.

Line 35, please consider capitalizing the name Gram, to honor its author.

Line 37, please consider replacing “Additionally”, by “In addition”.

Line 47, in my opinion it would increase the impact of the manuscript if the authors included some statistical values of the number of infections.

Line 95, please format the exponential value with super script.

Comments on the Quality of English Language

The authors should revise the term "additionally" in the begining of the sentences by "in addition".

That was the only issue that I found.

Reviewer 3 Report

Comments and Suggestions for Authors

The manuscript by Kim et al. provides the first complete genome of Dermacoccus barathri and its comparative analysis. 

I must acknowledge that the bioinformatical analysis is well done. The results are nicely and correctly visualized in figures and tables. The raw and assembled data is publically available. The results are reproducible. 

I recommend this paper for publication after minor corrections.

1. L. 17. Please, provide one sentence about the significance of the Dermacoccus barathri as the introduction of the abstract. The general reader is probably not familiar with this bacteria. The current introduction could be a little confusing.

2. L. 69-90. Please, consider writing the manufacturer's name and the country of origin of any equipment or kit used for the analysis.

3. L. 116-119. Please, consider visualizing the results of the BUSCO assessment as a figure or table. This could improve the overall appeal of the manuscript to the readers.

4. L. 120-121. This is mostly true for organisms with relatively short genomes. Please, consider rewriting this statement. In its current form, it can confuse the readers and make them think that Nanopore sequencing is the ultimate tool for de novo sequencing, which is not, at least at the moment. De novo sequencing of large genomes still requires high-accuracy reads (for example, from Illumina and PacBio HiFi).

5. L. 122. Please, consider adding the reference for the Homopolish tool.
